# Compound Capecitabine Colon-Targeted Microparticle Prepared by Coaxial Electrospray for Treatment of Colon Tumors

**DOI:** 10.3390/molecules27175690

**Published:** 2022-09-03

**Authors:** Ruiqi Chen, Ruidong Zhai, Chao Wang, Shulong Liang, Jing Wang, Zhepeng Liu, Wenlin Li

**Affiliations:** 1School of Health Science and Engineering, University of Shanghai for Science and Technology, Shanghai 200093, China; 2Department of Cell Biology, Second Military Medical University, Shanghai 200433, China; 3Shanghai WD Pharmaceutical Co., Ltd., Shanghai 201203, China

**Keywords:** coaxial electrospray, targeted delivery, combinatorial therapy, capecitabine, osimertinib, drug delivery

## Abstract

To improve the antitumor effect of combined capecitabine (CAP) and osimertinib (OSI) therapy and quickly and efficiently reduce tumor volumes for preoperative chemotherapy, we designed a compound CAP colon-targeted microparticle (COPMP) prepared by coaxial electrospray. COPMP is a core–shell microparticle composed of a Eudragit S100 outer layer and a CAP/OSI-loaded PLGA core. In this study, we characterized its size distribution, drug loading (DL), encapsulation efficiency (EE), differential scanning calorimetry (DSC), Fourier transform infrared spectra (FTIR), in vitro release, formula ratio, cellular growth inhibition, and in vivo antitumor efficacy. COPMP is of spherical appearance with a size of 1.87 ± 0.23 μm. The DLs of CAP and OSI are 4.93% and 4.95%, respectively. The DSC showed that the phase state of CAP and OSI changed after encapsulation. The FTIR results indicated good compatibility between the drug and excipients. The release curve showed that CAP and OSI were released in a certain ratio. They were barely released prior to 2 h (pH 1.0), less than 50% was released between 3 and 5 h (pH 6.8), and sustained release of up to 80% occurred between 6 and 48 h (pH 7.4). CAP and OSI demonstrated a synergistic effect on HCT-116 cells. In a colon tumor model, the tumor inhibition rate after oral administration of COPMP reached 94% within one week. All the data suggested that COPMP promotes the sustained release of CAP and OSI in the colon, which provides a preoperative chemotherapy scheme for the treatment of colon cancer.

## 1. Introduction

Colon cancer (CC) is the third-highest cause of cancer death worldwide, despite advances in treatment modalities, and the incidence rate of CC is increasing at an alarming speed in developing nations [1]. The proportion of CC in all diagnosed cancers and cancer-related deaths globally is reported to be 10% [2]. Because colon cancer involves multiple tumors, surgery alone cannot completely and effectively remove the cancer, except in the case of early tumors [3]. Conventional postoperative chemotherapy tends to result in drug resistance and systemic side effects [4,5]. In The Lancet Oncology, the Foxtrot Collaborative Group assessed the role of preoperative chemotherapy in treating locally advanced colon cancer. Their results demonstrated that local colon cancer is compatible with the toxicity of preoperative chemotherapy [6]. The potential advantages of preoperative chemotherapy include rapid reductions in the tumor volume and reduced likelihood the possibility of tumor cell proliferation before surgery, as well as killing distant tumor cells to reduce the possibility of recurrence, and enabling the determination of a postoperative chemotherapy scheme on the basis of preoperative chemotherapy effects [7,8]. In light of these characteristics, researchers are paying increasing attention to microparticles with targeting abilities to improve the therapeutic effect of preoperative chemotherapy [9].

In pursuit of this aim, the primary obstacle is the lack of selectivity between tumor cells and normal cells which results in inadequate drug concentration and drug resistance [10,11]. To avoid an inadequate drug concentration caused by systemic absorption, we may consider a directional sustained-release preparation to achieve colon-targeted administration [12,13]. For effective colon administration, the drug must have precise colonic targeting. It should either be readily absorbed by the colon or be specifically designed to treat colonic diseases. Oral colon-specific drug delivery systems (OCDDSs) can be designed to take advantage of pH changes and longer release times in the gastrointestinal tract. For example, Iurciuc et al. prepared curcumin-loaded complex particles (ComPs) based on three polysaccharides (gellan, i-carrageenan, and chitosan) obtained by ionic crosslinking and polyelectrolyte complexation. The ComPs were designed to allow efficient targeting and rapid curcumin release into the colon [14,15]. To address the problem of resistance caused by single-drug treatments, a multi-drug combination therapy can be designed. However, we must pay attention to the drug release ratio, toxic dose, and cooperation index. We should also bear in mind that the preparation of multi-drug particles is more difficult than that of single-drug particles.

Capecitabine (CAP), a pro-drug that is converted to 5-fluorouracil (5-FU) after oral administration, has become the first-line drug for preoperative chemotherapy of CC because it is readily absorbed from the gastrointestinal tract (GIT). However, treatment with CAP can result in the misincorporation of 5-FU metabolites into DNA, as well as impaired DNA replication, synthesis, and repair, eventually leading to DNA breaks [16]. The use of CAP alone has certain limitations in the treatment of colon cancer, including adverse side effects, drug resistance, and a short biological half-life of less than 1 h. Most importantly, it cannot efficiently reduce tumor volume [17,18]. To address these problems, osimertinib (OSI), an irreversible inhibitor against tyrosine kinase, is incorporated for the treatment of CC. The Lkb1/AMPK signaling pathway is essential for colorectal tumor cell survival and is a putative therapeutic target for CC [19]. As a protein tyrosine kinase, epidermal growth factor receptor (EGFR) is overexpressed in colorectal cancer [20]. OSI, an irreversible EGFR/HER2 inhibitor, can upregulate the expression of monocarboxylate transporter 1 (MCT1) and then activate LKB1/AMPK signaling, leading to autophagy induction in colorectal cancer cells [21,22]. Mechanistically, OSI increases the accumulations of other drugs in cells by inhibiting the efflux function of the transporters in ABCB1- or ABCG2-overexpressing cells [23]. Therefore, in this study, we used a combination of CAP and OSI to treat colon cancer, in the hope that OSI will overcome the drug resistance of CAP. However, when two drugs are selected to treat colon cancer, the therapeutic effect is likely to be unsatisfactory, unless the release ratio of the two drugs is considered [24].

Microparticles prepared by coaxial electrospray can be released in a certain proportion [25]. In this study, we used coaxial electrospray to prepare microparticles instead of traditional preparations [26]. With the help of coaxial electrospray, multilayer structures can be built by a one-step process, which is more convenient than conventional methods, and the encapsulation efficiency is higher [27]. Researchers have attempted to build shell particles by means of a coaxial electrospray. Hao et al. fabricated aspirin-loaded enteric-coated sustained-release nanoparticles with a core-shell structure using a coaxial electrospray [28]. Hitomi et al. synthesized AuNPs by liquid-phase electrospray under various nozzle potentials and temperatures [29].

To the best of our knowledge, this is the first report on using core–shell microparticles with CAP and OSI prepared by electrospray as an OCDDSs to suppress tumors in the short term for preoperative chemotherapy (Figure 1). We prepared a microparticle (COPMP) with two drugs (CAP and OSI) in the inner layer, and enteric coating in the outer layer. We then characterized the morphology, size distribution, drug loading, encapsulation efficiency, differential scanning calorimetry (DSC), Fourier transform infrared (FTIR) spectra, in vitro release, cellular growth inhibition, and in vivo antitumor efficacy.

## 2. Results

### 2.1. Formula Ratio

Table 1 shows that the IC_50_ values of free drugs were higher than those of the corresponding drugs in a free CAP/OSI physical mixture, regardless of the proportion of drugs. CAP and OSI exhibited a synergistic effect on HCT-116 cells. The IC_50_ of the free CAP/OSI physical mixture (1:1) was 0.1901 μM, much lower than that of free CAP (0.6990 μM) and OSI (0.6373 μM). We know that 0.9 ≤ CI_50_ ≤ 1.1 indicates superposition, 0.8 ≤ CI_50_ < 0.9 indicates low synergy, 0.6 ≤ CI_50_ < 0.8 indicates moderate synergy, and CI_50_ < 0.6 indicates high synergy [30]. These results also showed that when the drug ratio was 1:1, the strongest synergistic effect (CI_50_, 0.57) on HCT116 cells was recorded compared with the other ratios. For this reason, we used a COPMP drug ratio of 1:1 for subsequent in vivo and in vitro experiments.

### 2.2. Preparation and Characterization of Microparticles

Coaxial electrospray can encapsulate a drug in an inner solution to achieve sustained release [31,32,33]. To obtain the highest EE and the smallest particle size, we used a drug concentration of 0.5%. At this level of concentration, the DLs of CAP and OSI in the COPMP were 4.93% and 4.95%, respectively. The EE levels of CAP and OSI were 92.9% and 93.1%, respectively (Appendix A). As shown in Figure 2a-1,a-2, the particle sizes of COPMP and PMP were 2.11 ± 0.21 μm and 1.87 ± 0.23 μm, respectively, with this slight increase being the result of drug loading. Both particles exhibited a spherical morphology (Figure 2b-1,b-2). Researchers have often used thermal analysis to reveal possible changes in the physical state of drug carriers. As shown in Figure 2c, the thermograms of bulk CAP and OSI exhibit exothermic peaks at about 125 and 102 °C, respectively, and we observed no exothermic peaks at the same temperatures in COPMP, CPMP, or OPMP, indicating that the phase states of CAP and OSI changed after they were encapsulated in particles. The COPMP may have been in a solid solution state. Figure 2d presents the FTIR spectra of CAP, OSI, PMP, CPMP, OPMP, and COPMP. The spectra of the COPMP showed the characteristic absorption peaks of OH–/N–H stretching at 3519 cm^−1^ for CAP, and at 3435 cm^−1^ (v C = C) for OSI [34,35]. The observed peaks of PLGA in PMP, CPMP, OPMP, and COPMP at 1748 cm^−1^ occurred due to carbonyl stretching vibrations and the peak at 1165 cm^−1^ was due to ester stretching vibrations [36]. No new peaks formed in COPMP, indicating compatibility between drugs and excipients during coaxial electrospray.

### 2.3. In Vitro Drug Release

The in vitro behavior of CAP and OSI release from COPMP in buffer is shown in Figure 2e-1. The CAP and OSI exhibited a sustained release, indicating that the notion of COPMP releasing CAP and OSI in a certain ratio was successfully achieved. COPMP may play a role in the synergistic effect of the two drugs. We loaded the CAP and OSI into the core of the particles at the same time by electrospray, and CAP and OSI were sustainedly and proportionally released. CAP and OSI release was lower than 10% from 0 to 2 h, at pH 1.0 (gastric simulation); lower than 50% from 3 to 5 h, at pH 6.8 (small intestine simulation); and up to 80% for a sustained period of 6–48 h, at pH 7.4 (colon simulation). Figure 2e-2 shows that in the in vitro release behavior of the CPMP/OPMP physical mixture, OSI was released faster than CAP, except during the 4–6 h period. CAP and OSI were finally released at a level of 75%. Therefore, we inferred that the difference between the two release curves was caused by the formulation. Compared with simple physical mixing, compound microparticles can proportionally release the drug, thereby improving the efficacy of capecitabine, achieving the purpose of rapid tumor reduction before surgery and killing distant tumor cells to avoid recurrence.

### 2.4. Cellular Growth Inhibition

Figure 3 shows that the synergistic effect of COPMP was the strongest, with a CI_50_ of 0.42, while the CI_50_ of the free CAP/OSI physical mixture was 0.59, indicating that even if the free drugs were mixed, their toxicity to cells was still lower than that of COPMP. In the case of the CPMP/OPMP physical mixture, the CI_50_ was 0.56—higher than for COPMP, but lower than for the CAP/OSI physical mixture. This result may have been related to the release ratio of the two drugs.

### 2.5. In Vivo Antitumor Effects of Microparticles

We evaluated the antitumor efficacy of COPMP using an HCT-116 orthotopic colon cancer model in nude mice (Figure 4a). The tumor and tumor inhibition rates for each treatment group are shown in Figure 4b,c. Compared with the PMP, CAP, and OSI treatment groups, the COPMP treatment group showed the best antitumor effects. The tumor inhibition rate of the COPMP treatment group reached 94% within one week, indicating rapid tumor reduction and providing strong support for preoperative chemotherapy. The antitumor effects in the CPMP/OPMP physical mixture treatment group were not as good as in the COPMP treatment group, which was in line with the results of the cell experiment. Again, we confirmed the idea that the antitumor effect is related to the release ratio of the two drugs. We also observed red spots on the skin of nude mice in all free drug groups after the first administration. We suspected this was caused by drug toxicity; however, this situation did not occur in other groups (Appendix A). Finally, as illustrated in Figure 4d, we observed that the weight of mice in the free CAP, OSI, and CAP/OSI treatment groups stopped rising on day 24, while no such effect occurred in the other treatment groups, indicating that they had less acute toxicity.

## 3. Materials and Methods

### 3.1. Materials

We obtained PLGA (50:50, MW = 9000) from Jinan Daigang Biomaterial Co., Ltd. We acquired capecitabine (CAP) from Shouguang Jiasheng Chemical Co., Ltd. (Shouguang, China) We obtained osimertinib (OSI) from Cangzhou Enke Pharma-tech Co., Ltd. (Cangzhou, China), and the Eudragit S100 from Shanghai Chineway Pharma-tech Co., Ltd. (Shanghai, China) We purchased all the other chemicals from Sinopharm Reagent Co., Ltd. (Shanghai, China)

### 3.2. Fabrication of the Eudragit S100/CAP-OSI-PLGA Microparticles (COPMP), Eudragit S100/CAP-PLGA Microparticles (CPMP), Eudragit S100/OSI-PLGA Microparticles (OPMP), and Eudragit S100/PLGA Microparticles (PMP)

We prepared the Eudragit S100/CAP-OSI-PLGA microparticles (COPMP) by coaxial electrospray. We prepared the outer solutions by dissolving the Eudragit S100 (1.2 g) in 40 mL of ethanol/water mixture (9:1, *v*/*v*). The PLGA solution loading CAP and OSI was the inner solution. We prepared the inner solutions (drug concentration 0.1%, 0.3%, 0.5%, 0.7%, or 0.9%, *w*/*v*) by dissolving PLGA (240 mg), CAP (8, 24, 40, 56, or 72 mg), and OSI (8, 24, 40, 56, or 72 mg) in 8 mL of dichloromethane.

We poured the inner and outer solutions into plastic syringes connected to a coaxial needle. We pushed the liquid with the syringe pump. We set mass flow rates at 2.5 mL/h for the inner solution and 5 mL/h for the outer solution. We applied a voltage of 11 KV, and maintained an acceptance distance of 13 cm. We subjected Eudragit S100/CAP-PLGA microparticles (CPMPs) and Eudragit S100/OSI-PLGA microparticles (OPMPs) to the same process conditions and environment, except that the active pharmaceutical ingredients (APIs) of the inner solution were added only to CAP and OSI. We used a similar method in the case of the Eudragit S100/PLGA Microparticles (PMP), except that no API was added to the inner solution in this case.

### 3.3. Structural Morphology and Size Distribution

We observed the morphology of COPMP and PMP using scanning electron microscopy (SEM) (Phenom ProX, Phenom, Shanghai, China). We measured particle size and size distribution by means of dynamic light scattering (DLS) (Mastersizer 2000, Malvern, UK) using a solution of microparticles diluted with deliquated water.

### 3.4. Drug Loading (DL) and Encapsulation Efficiency (EE)

We accurately weighed the lyophilized COPMP, which was then dissolved in DMSO and then diluted to 6 mL with 50% methanol. We analyzed the drug loading (DL) and encapsulation efficiency (EE) of CAP and OSI using high-performance liquid chromatography (HPLC) (Waters 2695, equipped with a binary pump and PDA detector). We calculated DL and EE values using the following formulas:$$\mathrm{DL}\left(\%\right)=\frac{\mathrm{Weight}\;\mathrm{of}\;\mathrm{the}\;\mathrm{drug}\left(\mathrm{CAP}\;\mathrm{or}\;\mathrm{OSI}\right)\;\mathrm{in}\;\mathrm{microparticles}}{\mathrm{Weight}\;\mathrm{of}\;\mathrm{the}\;\mathrm{total}\;\mathrm{microparticles}}\times 100\%$$
$$\mathrm{EE}\left(\%\right)=\frac{\mathrm{Weight}\;\mathrm{of}\;\mathrm{the}\;\mathrm{encapsulated}\;\mathrm{CAP}\;\mathrm{or}\;\mathrm{OSI}}{\mathrm{Weight}\;\mathrm{of}\;\mathrm{the}\;\mathrm{total}\;\mathrm{CAP}\;\mathrm{or}\;\mathrm{OSI}}\times 100\%$$

The chromatographic apparatus consisted of an Agilent^®^ C18 column (25 cm × 4.6 mm with particle size of 5 μm; Agilent Technology, Shanghai, China). The mobile phase was a mixture of methanol and 3.25 mM sodium dihydrogen phosphate solution (we adjusted pH to 2.5 with phosphoric acid) at a ratio of 50 to 50 (*v*/*v*). The column temperature was 30 °C, the flow rate was 1.0 mL/min, and the detection UV wavelength was 213 nm. The retention times of CAP and OSI were approximately 14.8 and 7.8 min, respectively (Appendix A).

### 3.5. Physical Characterization

We characterized the final products, including the CAP, OSI, COPMP, CPMP, OPMP, and PMP, using a Fourier transform infrared spectrometer (FTIR, AVATAR 360, Nicolet, MA, USA) and a differential scanning calorimeter (DSC, Waltham, MA, USA). We set the FTIR to a 4 cm^−1^ resolution in a range of 450–4000 cm^−1^. DSC scanned from 25 to 280 °C with a heating rate of 10 °C/min and a nitrogen purge of 10 mL/min.

### 3.6. In Vitro Release

We applied the dialysis method to assess in vitro drug release. We placed 20 mg of microparticles (COPMP or CPMP/OPMP physical mixture) in 5 mL of hydrochloric acid buffer (0.1 mol/L) as the release medium, which we shook at 100 rpm at 37 °C for 2 h. After centrifugation, we removed the supernatant and put the precipitated particles into the dialysis bag (with a molecular weight cut-off value of 1000 Da). We then added 25 mL of PBS buffer solution (pH 6.8, containing 0.1% Tween 80) as the release medium and shook it at 100 rpm at 37 °C for 3 to 5 h. After further centrifugation, we again removed the supernatant and put the precipitated particles into the dialysis bag. We then added 25 mL of PBS buffer solution (pH 7.4, containing 0.1% Tween 80) as the release medium, which we shook at 100 rpm at 37 °C for 6 to 48 h.

### 3.7. Cytotoxicity Assay

#### 3.7.1. Formula Ratio

We measured the in vitro cytotoxicity of free CAP, free OSI, and the free CAP/OSI physical mixture with different drug ratios against HCT-116 cells using the standard MTT method. We cultured HCT-116 (5000/well) in a 96-well plate overnight. We incubated a series of concentrations of free CAP, free OSI, and free CAP/OSI physical mixture with different drug ratios with the cells for 48 h, and then added MTT. After 4 h, we added DMSO to the cells. We determined absorbance using a BioTek Power Wave XS reader (Power Wave XS, BioTek, Winooski, VT, USA) at 490 nm to calculate the percentages of cell viability. We calculated the IC_50_ value with GraphPad Prism 8 (La Jolla, CA, USA) using the following formula:$${\mathrm{CI}}_{50}=\frac{D_{\mathrm{CAP}}}{{\mathrm{IC}}_{50,\;\mathrm{CAP}}}+\frac{D_{\mathrm{OSI}}}{{\mathrm{IC}}_{50,\;\mathrm{OSI}}}$$
where IC_50, CAP_ and IC_50, OSI_ are the IC_50_ values of free CAP and free OSI when used alone, respectively, and D_CAP_ and D_OSI_ are the IC_50_ values of CAP and OSI when used together, respectively.

#### 3.7.2. Cellular Growth Inhibition

We measured the in vitro cytotoxicity of free CAP, free OSI, the free CAP/OSI physical mixture, PMP, COPMP, CPMP, OPMP, and the CPMP/OPMP physical mixture against HCT-116 cells using the standard MTT method. The subsequent experimental procedure was similar to that described in Section 3.7.1.

### 3.8. In Vivo Antitumor Study

To establish an animal model of colon cancer, we first randomly divided 36 male nude mice into 9 groups (*n* = 4). Then, after abdominal anesthesia (10 μL of tribromoethanol), we cut the abdominal cavity of each animal with surgical scissors, injected HCT-116 (1.25 × 10^6^/10 μL) cells into the colon wall, and intramuscularly injected 50 μL penicillin sodium solution. We allowed the mice to grow for 3 weeks. After this, we gavaged the nude mice every two days, three times in total. The treatment groups were as follows: control (normal saline), PMP, free CAP, free OSI, free CAP/OSI physical mixture, CPMP, OPMP, COPMP, and CPMP/OPMP physical mixture. The dosage was 10 mg/kg for the animals of all groups. We monitored the mouse’s weight every three days. We sacrificed the mice by cervical dislocation and then measured the tumor volume. We calculated the tumor inhibitory rate using the equation:$$\mathrm{Tumor}\;\mathrm{volume}\left({\mathrm{mm}}^{3}\right)=\frac{L\times W^{2}}{2}$$
$$\mathrm{Tumor}\;\mathrm{inhibitory}\;\mathrm{rate}\left(\%\right)=\frac{M_{B}-M_{T}}{M_{B}}\times 100\%$$
where L and W denote the length and weight of the tumor, respectively, and M_B_ and M_T_ represent average tumor volumes in the blank and treatment groups, respectively.

### 3.9. Statistical Analysis

Data are expressed as mean ± SD. Statistical analysis was conducted using a one-way analysis of variance (ANOVA) using the Graph Pad Prism 8 software. *p* values below 0.05 were considered statistically significant.

## 4. Conclusions

To summarize, we successfully prepared compound capecitabine colon-targeted microparticles by coaxial electrospray. We used a COPMP drug ratio of 1:1 for subsequent in vivo and in vitro experiments based on the strongest synergistic effect (CI50, 0.57) on HCT116 cells compared with the other ratios. The drug concentration was 0.5% to ensure the highest EE and the smallest particle size. Four microparticles (PMP, CPMP, OPMP, and COPMP) were prepared for characterization. Compared with PMP, the particle sizes of COPMP had a slight increase as a result of drug loading. The DSC showed the phase states of CAP and OSI changed after they were encapsulated in particles. The FTIR showed the characteristic absorption peaks of CAP, OSI, and PLGA in COPMP with no new peaks, indicating compatibility between drugs and excipients during coaxial electrospray. In addition, there were the peaks of PLGA in PMP; the peaks of CAP and PLGA in CPMP; the peaks of OSI and PLGA in OPMP.

Compared with CPMP/OPMP physical mixture, CAP and OSI in COPMP were sustainedly and proportionally released. Because of the ratio of the strongest synergistic effect, compared with CAP, OSI, CAP/OSI physical mixture, CPMP, OPMP, and CPMP/OPMP physical mixture, treatment with COPMP resulted in faster reductions in higher levels of cellular uptake, and stronger cellular growth inhibition. These results were consistent with the results of in vivo antitumor effects. The efficacy of COPMP against colon cancer demonstrates the potential of compound capecitabine colon-targeted microparticles prepared by coaxial electrospray for oral administration in the treatment of colon cancer.

## Figures and Tables

**Figure 1 molecules-27-05690-f001:**
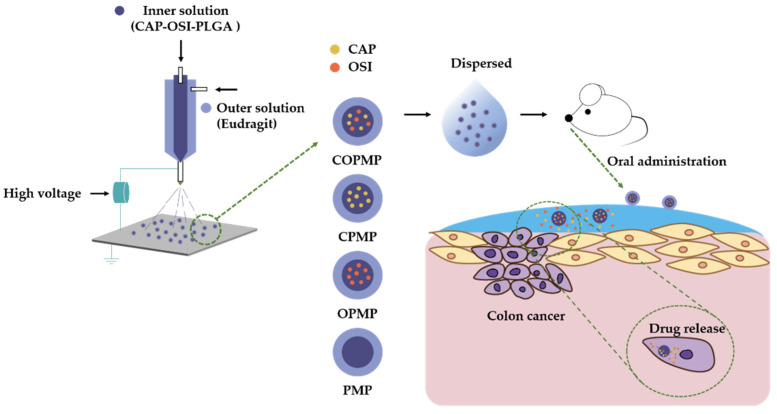
Illustration of COPMP: preparation of the core–shell microparticle by coaxial electrospray and treatment against colon cancer via oral administration. COPMP, CPMP, OPMP, and PMP refer to the inner layers of microparticles with CAP-OSI-PLGA, CAP-PLGA, OSI-PLGA, and PLGA alone, respectively, and likewise for the outer layers.

**Figure 2 molecules-27-05690-f002:**
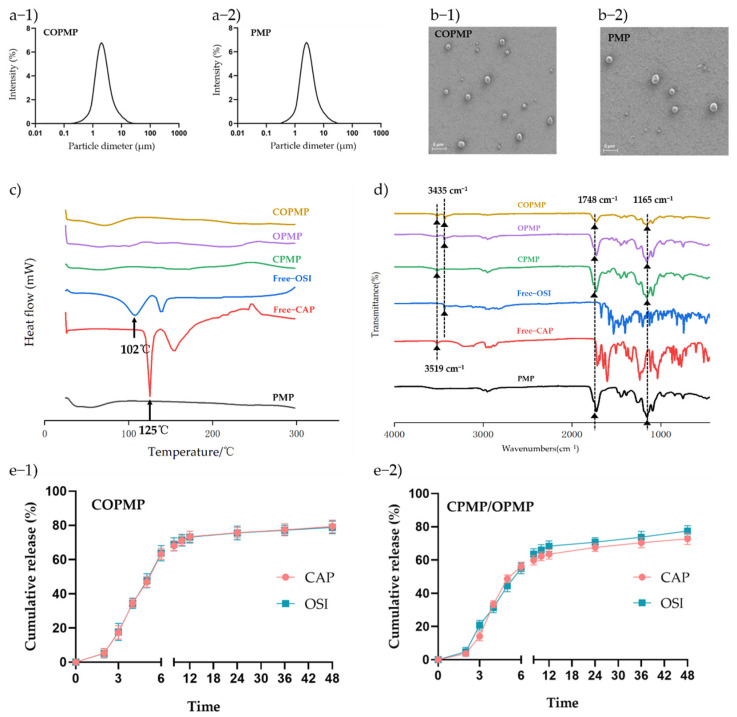
Characteristics of COPMP: (**a**-**1**) particle sizes of COPMP; (**a**-**2**) particle sizes of PMP; (**b**-**1**) SEM morphology of COPMP; (**b**-**2**) SEM morphology of PMP; (**c**) DSC; (**d**) FTIR; (**e**-**1**) in vitro release of CAP and OSI in COPMP; (**e**-**2**) in vitro release of CAP and OSI in CPMP/OPMP physical mixture (*n* = 3, mean ± SD).

**Figure 3 molecules-27-05690-f003:**
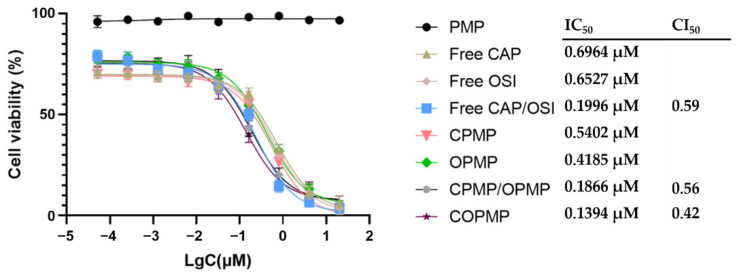
The IC_50_ of different administration groups (*n* = 3, mean ± SD).

**Figure 4 molecules-27-05690-f004:**
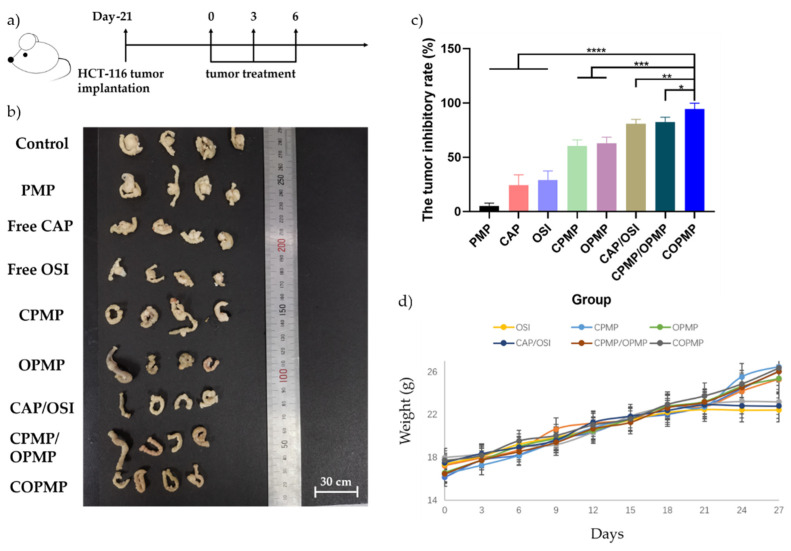
Therapeutic antitumor efficacy in an HCT-116 tumor model. (**a**) Schematic timeline of the efficacy study. Tumors were treated with normal saline, PMP, free CAP, free OSI, CPMP, OPMP, CAP/OSI, CPMP/OPMP, or COPMP (*n* = 4, mean ± SD); (**b**) tumor sizes for each group after treatment; (**c**) tumor inhibitory rates for each group after treatment; (**d**) body weight of mice after treatment (*n* = 4, mean ± SD). * *p* < 0.05, ** *p* < 0.01, *** *p* < 0.001, **** *p* < 0.0001.

**Table 1 molecules-27-05690-t001:** IC_50_ values of free CAP, free OSI, and CAP/OSI physical mixture with different drug ratios (*n* = 3).

CAP:OSI	IC_50,CAP_ (μM)	IC_50,OSI_ (μM)	CI_50_
Free CAP	0.6990		
CAP:OSI = 5:1	0.1651	0.3314	0.76
CAP:OSI = 2:1	0.1771	0.2851	0.70
CAP:OSI = 1:1	0.1901	0.1901	0.57
CAP:OSI = 1:2	0.3257	0.1643	0.72
CAP:OSI = 1:5	0.3571	0.1433	0.74
Free OSI		0.6373	

## Data Availability

Not applicable.

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
