# Peer review of "Compound Capecitabine Colon-Targeted Microparticle Prepared by Coaxial Electrospray for Treatment of Colon Tumors"

_molecules, 2022, doi:10.3390/molecules27175690_

Round 1
Reviewer 1 Report
This manuscript titled "Compound Capecitabine Colon Target Microparticle Prepared 2 by Coaxial Electrospray for Treatment of Colon Tumors" by Chen et al reports the investigatation of whether CAP 16 and OSI enteric-coated microparticle (COPMP) prepared by coaxial electrospray elicits antitumor 17 responses by oral administration.
This manuscript is well-written and scientifically sounds goods. This can be accepted without further modifications.
Reviewer 2 Report
The paper submitted by Chen et al. deals with the preparation of some dual drug-loaded microparticles by coaxial electrospraying for potential treatment of the colon tumors. The manuscript is interesting but the language is very poor and it needs to be spell and grammar checked by a native English speaking.
In order to increase the overall quality of the paper, the following corrections must be addressed:
1. the first sentence from the abstract must be rewritten.
2. the introduction section must be completed with some details about the requirements that a DDS must fulfill for an efficient colon administration. Moreover, the authors must cite the existing literature data about the different types of micro/nanoparticles suitable for colon administration. Some suggestions are: https://doi.org/10.1016/j.ijbiomac.2019.12.247; https://doi.org/10.3390/ijms22063075
3. L116: what is a “solid solution state”?
4. concerning the FTIR results, I would like to know if the authors have noticed some shifts of the characteristic peaks of PMP, CAP and OSI in COPMP? They have only discussed the absence of new peaks but it is very important to see if the characteristic peaks of the components have shifted as this is a clear indication of the existence of different types of interactions between the drugs and the polymer matrix.
5. if I understood properly, the results provided in subsections 2.1 and 2.2 are for drugs ratio of 1:1. If this is the case, then the subsection 2.3 must be moved at the beginning of the results section. Moreover, a reference must be added at the end of the sentence: “it is know that 0.9<CI50<1.1…is high energy.”
6. can the authors explain why the data concerning the CI50 given in fig 3 are different from those calculated in table 1?
7. section 3.4: the authors must explain how they destroyed the microparticles in order to recover of the quantity of the encapsulated drugs.
8. the conclusions section is very poor and must be completed. Only capecitabine-loaded microparticles were studied? There are a lot of grammatical errors in this section.
9. the full name of GIT and API must be provided.
Round 2
Reviewer 2 Report
The paper can be accepted as it is.
Author Response
We really appreciate your time and effort in handling reviews of this manuscript. We appreciated the encouraging comments of this reviewer.